The neglected bee trees: European beech forests as a home for feral honey bee colonies

http://orcid.org/0000-0001-9278-978X Kohl Patrick Laurenz 1
http://orcid.org/0000-0001-6589-6408 Rutschmann Benjamin 1 2 benjamin.rutschmann@uni-wuerzburg.de
1 Department of Animal Ecology and Tropical Biology, Biocenter, University of Würzburg , Würzburg , Germany
2 HOBOS, University of Würzburg , Würzburg , Germany
Negri Ilaria
Electronic publication date: 2018 Apr 6
Publication date: 2018
Volume: 6
Electronic Location ID: e4602
Received 2018 Feb 12; Accepted 2018 Mar 21
Copyright: © 2018 Kohl & Rutschmann
Copyright year: 2018
Copyright holder: Kohl & Rutschmann
License: This is an open access article distributed under the terms of the Creative Commons Attribution License, which permits unrestricted use, distribution, reproduction and adaptation in any medium and for any purpose provided that it is properly attributed. For attribution, the original author(s), title, publication source (PeerJ) and either DOI or URL of the article must be cited.
License URL: https://creativecommons.org/licenses/by/4.0/

Keywords: Feral honey bees, Wild honey bees, Apis mellifera, Dispersal, Tree cavity, Black woodpecker, Hollow tree, Swarming, Beech forests, Fagus sylvatica

Funding: German Research Foundation University of Wuerzburg This publication was funded by the German Research Foundation (DFG) and the University of Wuerzburg in the funding programme Open Access Publishing. The funders had no role in study design, data collection and analysis, decision to publish, or preparation of the manuscript.

==============================
It is a common belief that feral honey bee colonies (Apis mellifera L.) were eradicated in Europe through the loss of habitats, domestication by man and spread of pathogens and parasites. Interestingly, no scientific data are available, neither about the past nor the present status of naturally nesting honeybee colonies. We expected near-natural beech (Fagus sylvatica L.) forests to provide enough suitable nest sites to be a home for feral honey bee colonies in Europe. Here, we made a first assessment of their occurrence and density in two German woodland areas based on two methods, the tracing of nest sites based on forager flight routes (beelining technique), and the direct inspection of potential cavity trees. Further, we established experimental swarms at forest edges and decoded dances for nest sites performed by scout bees in order to study how far swarms from beekeeper-managed hives would potentially move into a forest. We found that feral honey bee colonies regularly inhabit tree cavities in near-natural beech forests at densities of at least 0.11–0.14 colonies/km2. Colonies were not confined to the forest edges; they were also living deep inside the forests. We estimated a median distance of 2,600 m from the bee trees to the next apiaries, while scout bees in experimental swarms communicated nest sites in close distances (median: 470 m). We extrapolate that there are several thousand feral honey bee colonies in German woodlands. These have to be taken in account when assessing the role of forest areas in providing pollination services to the surrounding land, and their occurrence has implications for the species’ perception among researchers, beekeepers and conservationists. This study provides a starting point for investigating the life-histories and the ecological interactions of honey bees in temperate European forest environments.

Introduction

The habitat of the western honey bee (Apis mellifera L.) in temperate Europe was deciduous or mixed forest, with trees providing both nest cavities and food resources (Seeley & Morse, 1976; Seeley, 1985; Ruttner, 1988; Crane, 1999). However, since the Middle Ages, the life of honey bees in Central Europe has considerably been affected by man. When people started to take colonies to their homes and manage them in beekeeping hives, the habitat of honey bees shifted from woodland areas to the agro-urban space (Crane, 1999; Banaszak, 2009). The logging of old-grown forests and an increase in land-use intensity further reduced the habitat for wild honey bee colonies (De la Rúa et al., 2009; Küster, 2010; Potts et al., 2010). Advances in beekeeping technique allowed for a better control over honey bee reproduction, altering their life-history and limiting natural selection (Loftus, Smith & Seeley, 2016; Neumann & Blacquière, 2017; Brosi et al., 2017). Lastly, the trade in selected honey bee strains by bee breeders across Europe has led to genetic admixture and spread of pathogens and invasive parasites (De la Rúa et al., 2009; Mutinelli, 2011) challenging the health of these important pollinators (Neumann & Carreck, 2010; Meixner, Kryger & Costa, 2015). Today, among the most harmful pests is the ectoparasitic mite Varroa destructor (Anderson & Trueman, 2000), which is infecting virtually all honey bee colonies across Europe (Le Conte, Ellis & Ritter, 2010; Rosenkranz, Aumeier & Ziegelmann, 2010; Nazzi & Le Conte, 2016). There is consensus among researchers and beekeepers that colonies not treated against the mite will usually die within a few years due to both direct damage and associated virus infections (Sammataro, Gerson & Needham, 2000; Moritz et al., 2010; Neumann & Carreck, 2010; Rosenkranz, Aumeier & Ziegelmann, 2010; Martin et al., 2012; van Dooremalen et al., 2012). The question is: what does that mean for feral honey bee colonies, which nest naturally with no human interference? It is widely assumed that since the arrival of the mite, feral honey bee colonies were all but wiped out (Moritz et al., 2007; De la Rúa et al., 2009; Meixner, Kryger & Costa, 2015). However, it has been reported that naturally nesting honey bees colonies are still abundant in large temperate woodlands in North America and the Southern Ural (Seeley, 2007; Ilyasov et al., 2015), and they regularly occupy cavity trees along rural avenues in Northern Poland (Oleksa, Gawroński & Tofilski, 2013). There are two reasons why we should reconsider the common perception about the status of honey bee colonies living wildly in Central Europe: different host-parasite ecology, and a simple lack of data.

In predicting the fate of feral colonies, we usually consult knowledge about honey bee-parasite relationships that was gathered in the context of managed apiaries. It is long recognized, however, that feral colonies generally experience a reduced parasite pressure due to ecological factors (Bailey, 1958). The crowding of honey bee colonies at bee yards increases the risk of erroneous homing by workers (drifting), which boosts epidemic spread among hives and individual colonies’ parasite loads (Fries & Camazine, 2001; Forfert et al., 2015; Seeley & Smith, 2015). Parasite loads were also found to be positively correlated with the colony density at the landscape scale (Frey & Rosenkranz, 2014; Forfert et al., 2016; Nolan & Delaplane, 2017). Because people tend to have their bee yards near settlements, beekeeping activity is not distributed evenly across space, resulting in high local colony densities. A study on parasite pressure of honey bees along an urbanization and management gradient in the Eastern United States, indeed found that parasite burden was both higher in urban as compared to rural locations, and in managed as compared to feral hives (Youngsteadt et al., 2015). In contrast, worker bees of feral colonies in the UK had higher virus infestations than workers of managed colonies (Thompson et al., 2014). However, the sampling was potentially biased towards locations in urban areas with high overall colony densities. The density of naturally nesting colonies in remote woodlands is low and colonies nest individually in widely separated tree cavities (Visscher & Seeley, 1982; Seeley et al., 2015; Seeley, 2017). In addition, the life-history strategy of feral colonies makes them less vulnerable to parasites than beekeepers’ colonies. Naturally nesting colonies stay smaller, rear less brood and swarm frequently, all of which reduces the reproductive potential of Varroa mites. Beekeepers, however, usually prevent swarming and provide unnaturally spacious hives, resulting in large colonies with continuous brood rearing activity (Loftus, Smith & Seeley, 2016; Seeley, 2017). Both empirical and theoretical studies suggest that these factors alone—the spacing of colonies, and their life-history—can reduce the infection levels of V. destructor considerably (Loftus, Smith & Seeley, 2016; DeGrandi-Hoffman, Ahumada & Graham, 2017; Seeley, 2017). Further, it is predicted that under certain conditions, populations of honey bee colonies not treated against parasites will evolve resistance mechanisms against these pests through natural selection (Neumann & Blacquière, 2017). Selection for resistance against Varroa was indeed documented in several studies of both closed and open managed populations that did not receive treatment against the mites (Rinderer et al., 2001; Le Conte et al., 2007; Locke & Fries, 2011; Locke et al., 2012; Kefuss et al., 2015; Locke, 2016; Oddie, Dahle & Neumann, 2017). So far, however, only one population of feral colonies has been studied with respect to its survival and host-parasite ecology: although the new parasite and its associated viruses evidently led to an initial drop in the population sizes of feral honey bee colonies in North America (Kraus & Page, 1995; Mikheyev et al., 2015), a population of honey bees living in the deciduous forests of New York State was found to be stable over decades (Seeley, 2007; Mikheyev et al., 2015; Seeley et al., 2015; Seeley, 2017). This population exhibits genomic signatures of selection after the arrival of Varroa (Mikheyev et al., 2015) and individual bees were found to show intense hygenic and biting behavior against the mites (D. Peck & T. D. Seeley, 2018, personal communication). Interestingly however, a high winter mortality of founder colonies in new nests suggests that the population is more limited by food supply than by parasites (Seeley, 1978, 1985, 2017).

The fact that we owe our knowledge about forest dwelling temperate honey bees almost exclusively to a feral population outside its natural range, from the deciduous forests of New York State (Seeley, 1985), points to the second problem: there are actually no scientific data available about the ecology of feral honey bee colonies in Europe, neither before nor after the arrival of the Varroa mite (Geldmann & González-Varo, 2018). It is often quoted that feral honey bee colonies have always been rare (Ruttner, 1973), but there is no information about their actual abundance and distribution at earlier times. Further, the density of honey bee colonies in temperate woodlands might naturally be very low and nests are extremely cryptic (Seeley & Morse, 1976; Visscher & Seeley, 1989): discovering a honey bee nest by chance and without using specific searching techniques is hence very unlikely. To overcome the difficulty in directly detecting honey bee nests, genotyping of sampled drones or queen offspring with microsatellite markers was used to estimate honey bee colony densities in different regions across their native range (Moritz et al., 2007; Moritz, Dietemann & Crewe, 2008; Jaffé et al., 2010). At sample sites in Europe, inferred colony densities resembled those of known managed colonies, so it was concluded that wild or feral honey bees do not contribute to the population as a whole (Moritz et al., 2007; Jaffé et al., 2010). However, it is probable that the resolution of the method is not precise enough to detect low densities of feral colonies in regions where many more colonies live in managed hives (Arundel, Oldroyd & Winter, 2012, 2013).

Getting precise information about the presence of feral honey bee colonies is important: they need to be considered when estimating overall wild pollinator abundances or when assessing the role of forest areas in providing pollination services to the surrounding land (Tscharntke et al., 2005; Jaffé et al., 2010; Mitchell, Bennett & Gonzalez, 2014; Decocq et al., 2016). The occurrence of feral honey bee colonies in woodlands would also have special implications for the species itself, for example its legal protection status and its perception among bee researchers, forest ecologists and conservationists. Although feral honey bee colonies can be a rich source for studying the natural interaction of honey bees with the forest environment (Seeley, 1985) and they can represent an important reservoir of genetic diversity (Oleksa, Gawroński & Tofilski, 2013), so far, they have got little attention in Europe.

In most parts of Central Europe, beech dominated forests (Fagus sylvatica L.) represent the major type of natural vegetation (Bohn et al., 2004; Tinner & Lotter, 2006; Bolte, Czajkowski & Kompa, 2007), so that increasing efforts are being devoted to the assessment of biodiversity in these forests and to their conservation (Brunet, Fritz & Richnau, 2010; Scherfose et al., 2007). Little is known about the role of beech forests as a home for pollinating insects (Sobek et al., 2009). At first glance, they actually appear to provide limited nectar resources: beech trees are wind pollinated, and mature stands form dense canopies, which limit light transmission to the forest floor and hence the growth of shrubs and other tree species (Packham et al., 2012). However, the wild bee fauna of the Hainich, an ancient beech forest in central Germany, was found to be quite diverse (Sobek et al., 2009). Further, there are historical reports about people hunting for honey in the Carpathians, a region where today’s last primeval beech forests are situated (Gunda, 1968), and honey bees were repeatedly reported to occupy old black woodpecker cavities (Dryocopus martius L.) in beech trees (Meyer & Meyer, 2001; Sikora, 2008; Sikora, Schnitt & Kinser, 2016).

We assumed that near-natural deciduous forests in Germany could provide enough suitable nest sites to be a home for feral honey bee colonies. Here, we made a first assessment of their occurrence and density by using two methods, the tracing of nest sites based on forager flight routes (beelining technique), and the direct inspection of potential cavity trees. Further, we established experimental swarms at forest edges and decoded dances for nest sites performed by scout bees in order to study how far swarms from beekeeper-managed hives would potentially move into a forest.

Methods

Study sites

We conducted our study in two different beech forest areas in Germany, the Hainich and the forests of the Swabian Alb.

Hainich. The Hainich forest in central Thuringia (51.08, 10.43) is one of the largest non-fragmented stretches of deciduous forest in Central Europe. It lies at about 400 m a.s.l. and its climate is characterized by a mean annual temperature of 7.5 °C and a mean annual precipitation of 630 mm (Gauer & Aldinger, 2005). Since 1964, its southern part served as a military training ground, and was therefore not disturbed much over the last 60 years. Seven thousand six hundred ha from a total wooded area of 16,000 ha were designated National Park in 1997 and became a World Heritage Site as part of the primeval beech forests of the Carpathians and other regions in Europe (Knapp, Nickel & Plachter, 2007). The dominant tree species are beech (F. sylvatica L., 65%), ash (Fraxinus excelsior L., 25%) and maple (Acer pseudoplatanus L., Acer platanoides L. and Acer campestre L., 7%) in an uneven age distribution (1–250 years) (Kutsch et al., 2010). Beside these, there are also lime trees (Tilia cordata Mill. and Tilia platyphyllos Scop.) which, like maple trees, are rich nectar and pollen sources for honey bees and other pollinating insects (Tofilski & Oleksa, 2013). Due to the regulations of the authorities it is not allowed to keep managed honey bee colonies inside the National Park.

Swabian Alb. The Biosphere Reserve Swabian Alb is a characteristic, low-mountain cultural landscape in the southwest of Germany (48.41, 9.54). While the foothills at about 300 m a.s.l. have a mild climate with an annual mean temperature of 8 °C, the Alb plateau at about 800 m a.s.l. is considerably cooler with an annual mean temperature of 6.7 °C. Precipitation amounts to 750–1,050 mm annually. The Biosphere Reserve, which encompasses 85,000 ha around the former army training ground Münsingen, was formed in 2008 and designated as UNESCO Biosphere Reserve in May 2009. Large areas of the reserve consist of extensively used species-rich meadows, juniper heaths and fruit orchards, but there are also richly structured forests with many hollow trees that serve as shelter for birds, small mammals, bats and insects. The dominating tree species is beech (F. sylvatica L.), but ash (F. excelsior L.), maple (A. pseudoplatanus L.), elm (Ulmus glabra Huds.), lime (T. cordata Mill. and T. platyphyllos Scop.), hornbeam (Carpinus betulus L.), fir (Picea abies (L.), H. Karst) and spruce (Abies alba Mill.) are also common (Jooß, 2013).

Occurrence and density of feral honey bee colonies in the Hainich forest

The fact that foraging honey bees have high site fidelity and recruit nest mates to rich food sources can be used to trace feral colonies when their nest sites are difficult to find (Visscher & Seeley, 1989; Wenner, Alcock & Meade, 1992; Seeley, 2016). “Beelining,” the tracking of honey bees based on their flight paths is actually an ancient method, which has been tested and proven by honey hunters for centuries (Gunda, 1968; Crane, 1999).

In September 2016, June 2017 and September 2017, we spent a total of 10 days searching for feral honey bee colonies in the Hainich forest. We worked during dearth seasons, when nectar was scarce and honey bees readily accepted our artificial feeding stations. On meadows inside or near the Hainich forest, we caught foraging honey bees with a special “beelining box.” Inside this dark box, we fed the bees highly concentrated sugar syrup (approximately 2.0 mol/l) flavored with anise essence. After release, most foragers oriented themselves, flew away and came back some minutes later. By means of different shellac color paints, we individually marked the bees and recorded their round trip times. Bees of the same nest can vary considerably in round trip duration, because they have different propensities to dance and spend different amounts of time inside the nest (Arnold et al., 2002). Therefore, to estimate the maximum distance of the bee tree, we considered the trip duration of the fastest bees. After a few round trips, the bees usually use an almost direct flight path (the “beeline”) between the feeder and their nest, so that their flight direction can be recorded. To calculate maximum nest distances, we used the following empirical formula which accounts for the homeward flight, the landing and unloading at the nest, and the rushing back and landing at the feeder: distance (m) = 243 × t (min) − 627.75, where t is the round trip time of the fastest bee (Seeley, 2016). Based on a pilot beelining test in a forest area near Würzburg, we knew that the formula was generally suitable in our terrain. We used the average flight direction and the inferred maximum nest distance to make a prediction of the nest location. Then we trapped a bunch of foragers in the box again and released them at a new feeding site a couple of hundred meters in the direction of the nest. We repeated this procedure until the fastest bees only needed around three minutes for a round trip, which is when the bee tree is no more than 100 m away (Seeley, 2016). We then started searching the trees in the area for the nest cavity with binoculars. In order to get information about the locations of apiaries around the Hainich National Park, we spoke to local beekeepers. To estimate the minimum density of feral colonies in the Hainich forest, we assumed that by catching foraging bees at a site with flowers in bloom, we would have obtained information about nests in an area of about 2 km around that site, as this corresponds to the area honey bee colonies usually forage (Visscher & Seeley, 1982; Steffan-Dewenter & Kuhn, 2003; Couvillon et al., 2015; Danner et al., 2016). We sampled slightly different but overlapping areas in the two years, an estimated 30.58 and 43.23 km2 in 2016 and 2017 respectively, totaling 49.50 km2.

Occupation of old woodpecker cavities by honey bee colonies in the Swabian Alb

Beelining is a useful method for exploring the general occurrence and density of feral colonies in a woodland from scratch. However, it can be difficult to exactly localize the bees’ nesting sites, a prerequisite for many ecological studies. An alternative way to determine the locations of feral honey bee colonies is to directly inspect suitable cavity trees in the region of interest (Oleksa, Gawroński & Tofilski, 2013). A cavity for honey bees needs a volume of at least 20 l in order to allow them to hoard enough honey to overwinter (Seeley, 1985). The black woodpecker (D. martius L.) is the largest woodpecker in the Palearctic and excavates spacious nest cavities (Johnsson, Nilsson & Tjernberg, 1993; Blume, 1996), making it a key candidate for creating nest sites that are suitable for honey bee colonies in managed forests. In the forest clusters of the Biosphere Reserve Swabian Alb, there is a long record of trees worked by black woodpeckers (Sikora, Schnitt & Kinser, 2016). Because of the high ecological value of microhabitat structures in managed beech forests, such trees get permanently marked and are protected as a measure of biodiversity promotion (Bütler et al., 2013). In September 2017, we inspected 98 trees from a list of 282 trees in a forest area of 180 km2 that are known to contain woodpecker cavities older than 10 years (Sikora, Schnitt & Kinser, 2016). The selection of sites was based on logistic considerations. We found the tree locations with the aid of GPS coordinates. We inspected cavities with binoculars from the ground during weather conditions suitable for honey bee foraging. A cavity was considered occupied by a feral colony when there was regular flight traffic of foraging worker honey bees and pollen import. To make an estimate of the colony density, we calculated the density of all 282 habitat trees in the beech forest clusters and multiplied it with the occupation rate of the inspected cavities in fall 2017. Note that this density estimate must be regarded as a lower limit because we only considered honey bee colonies nesting in old woodpecker cavities.

Potential dispersal distances of honey bee swarms taking off from apiaries near the Hainich forest

In the context of honey bees colonizing woodlands, an interesting question is how far swarms from beekeeper-managed colonies would usually move into the forests. When swarming bees leave their old nest, they first settle in close vicinity and form a hanging swarm cluster, typically on a tree branch. Scout bees then search the area for suitable nesting sites and report the location of their finds back at the swarm cluster via the waggle dance. Other bees that attend these dances may fly to the advertised sites, and—if they consider one appropriate—join in the dancing. Individual bees cease dancing after some time, regardless of the quality of the advertised site, so information about the best nesting sites accumulates over time through tradition, while information about inferior sites gets gradually lost. After hours or days, the outcome is a consensus about where to go, which is followed by the swarm’s lift-off and move towards the new home (Lindauer, 1955; Seeley, 2010). As scout bees report their findings publicly on the swarm cluster, a human observer can easily obtain information about a swarm’s potential nesting sites by observing and reading the bees’ dances (Lindauer, 1955; Seeley & Morse, 1977).

In August 2017, we investigated the potential dispersal behavior of swarms taking off near the Hainich forest, by decoding the nest site dances of three experimental swarms set up around the National Park. To prepare swarms, we captured the queen of a source colony and put her into a small cage of meshwork. Then, we swept 850–950 g worker bees (7,000–8,000 bees) off their combs into a screen cage. Together with their queens, the swarms were kept in a cool and dark place for >72 h and fed 50% sucrose solution ad libitum, provided through a gravity feeder bottle. It is known that artificial swarms prepared like this behave like natural ones in that they start searching for nest sites and eventually move to a new home (Morse & Boch, 1971; Seeley & Morse, 1977). To simulate realistic dispersal events from managed hives, we set up the swarms at sites around the Hainich forest, which were near settlements or actually close to an apiary: in Weberstedt (northeast of the Hainich, 10.49, 51.11, near an apiary, distance to forest: 870 m), in Craula (southeast of the Hainich, 10.47, 51.06, distance to forest: 450 m), and at Mallinde (southwest of the Hainich, 10.40, 51.05, distance to forest: 170 m). We used a board of plywood (50 × 45 cm) vertically fixed to a wooden stand as a swarm observation mount. We positioned the caged queen at a gap in the center of the board and let the workers out of the screen cage so that they could assemble on the board around their queen. When established, we started video recording the swarms for a later analysis of nest site dances. In order to avoid having bees dancing for nectar sources, we continued to feed the swarms sucrose during the experiments (Seeley & Buhrman, 1999). We worked during optimal weather conditions for swarming, on a sunny day with temperatures >20 °C. As the swarms’ queens were confined to a cage at any time, we prevented the swarms from actually moving to a new nest cavity. Thus, scout bees could search for nest sites, dance and reach a consensus about where to go, but lifted swarms would always have to come back to the swarm mount and their queens. On all three swarms we observed dances soon after the setup was established, and two swarms (in Craula and in Weberstedt) actually tried to move after some hours. In the evening after the observation, we swept the bees off the swarm mount into the screen cage again, and brought the colonies back to their original hives.

For decoding the nest site dances, we played the videos until a dance commenced. Then we measured the duration of around six consecutive dance circuits per dance to calculate an average circuit duration for each dance and aligned a digital protractor to obtain the dance angle in relation to the vertical. We translated dance circuit duration into nest site distance based on the original data of Von Frisch (1967). The distance-dance circuit duration relationship is well described by two linear regression equations, with a breaking point at about 1,000 m distance. We used distance (m) = 466.5495 × circuit duration (s) − 675.0336 for circuit durations up to 3.15 s, and distance (m) = 1102.7328 × circuit duration (s) − 2666.6256 for circuit durations >3.15 s (see Fig. S1). To get to know the direction of the advertised nest sites, we calculated the solar azimuth at the time of each dance using the AstroWin 32 software (Strickling, 2012) and added the solar azimuth angles to the dance angles. We could not decode all dances performed by the scout bees, as these did not dance exclusively on the surface of the swarm. Some opportunistically used the wooden board as a dance floor and communicated nest sites covered by the swarm cluster out of our sight. However, as the observed dances were most likely a random portion of all dances, they should well represent the overall range of nest site distances and preferences by the swarms.

Results

Occurrence and density of feral honey bee colonies in beech forests

In total, we detected nine bee trees in the Hainich forest (Fig. 1). All nest sites were located in areas with dense tree canopies. This hampered the determination of exact flight directions near the nest sites and locating the actual cavity tree was infeasible. We therefore determined the approximate location of each bee tree by taking into account the round trip durations of bees foraging at the last feeder and the information obtained at the previous feeding sites (for detailed examples of how we inferred the bee tree locations see Fig. S2). In 2016, we inferred the bee tree locations A, B, C and D. In 2017, we sampled a slightly different area and detected bee tree location D again plus the bee tree locations E, F, G, H and I. This corresponds to an estimated average colony density of 0.13 colonies per square kilometer (0.13 col/km2 in 2016, 0.14 col/km2 in 2017). While five bee trees lay within the core zone of the National Park, another two lay within its boundaries, and two just outside the boundaries of the National Park. We got to know that in every village around the National Park there is at least one apiary with around 10 managed colonies each. In one case, the inferred location of a potential bee tree (location H) was near the forest edge, so we specifically searched that area for beekeepers’ hives to be sure that the tracked bees came from a feral colony. The approximate mean ± SD distances of the bee trees to the forest border and to the nearest apiary were 980 ± 830 and 2,660 ± 1,080 m respectively (Table 1).

Figure 1 Map of the bee tree locations inferred from beelining data in the Hainich forest in the years 2016 and 2017.

Arrows represent recorded beelines (putative flight vectors of foragers from artficial feeders to their nests), with an arrow tip predicting a bee tree location. The vector lengths give the maximum distances of the nests based on round trip times of the fastest bees, and the vector angles give the direction of the nest based on average flight directions. Dashed arrows represent beelines to bee trees or apiaries that probably lay outside of the map because the respective bees had long round trip durations. Note, however, that we might have overestimated nest distances when the observed bees spent long times inside their nests between foraging flights (e.g., due to recruitment dancing activity) (Germany map created in R with package “maps” (R Core Team, 2016; Brownrigg, 2017). Other map data © Mapbox, © OpenStreetMap).

Table 1 Distances of the inferred bee tree locations in the Hainich forest to the forest edge and to the next apiary.

Bee tree	Distance to forest edge (m)	Distance to next apiary (m)	
A	2,350	4,300	
B	700	2,600	
C	1,700	3,600	
D	2,250	4,000	
E	550	2,850	
F	150	1,500	
G	500	1,500	
H	<50	1,150	
I	550	2,400	
Median	550	2,600	
Mean	980	2,660	
SD	830	1,080	

In the beech forests of the Biosphere Reserve Swabian Alb we inspected 98 habitat trees with old black woodpecker cavities for the presence of honey bee colonies (see Fig. S3 in supplemental information for a map of the region). Many of the habitat trees contained more than one cavity (Fig. 2). Seven trees (7.1%) were found to be occupied by honey bees. Based on the density of mapped woodpecker cavity trees in the region, we estimated the density of feral honey bees to be at least 0.11 colonies per square kilometer.

Figure 2 Bee trees in the Swabian Alb.

(A) Portrait of a bee tree with three cavity openings made by the black woodpecker. (B) close-up of an old woodpecker cavity occupied by a honey bee colony. Both trees are beech (Fagus sylvatica L.) (Germany map created in R with package “maps” (R Core Team, 2016; Brownrigg, 2017). Photo credit: Patrick Laurenz Kohl).

Potential dispersal distances of honey bee swarms

Nest site scouts of three experimental swarms set up around the Hainich forest apparently covered a large area, dancing for nest sites up to 6.5 km away (Fig. 3). However, the majority of dances indicated sites within a few hundred meters around the swarm mounts. Bees of the swarms in Craula and at Mallinde communicated nest sites located in the Hainich forest, but those of the swarm in Weberstedt did not. Potential nest site hot spots (sites which were advertised repeatedly by scout bees) of the swarms in Weberstedt and Craula lay outside the forest, while the swarm at Mallinde showed preferences for nest site at the forest margin. Average median and 95-percentile nest site range for the three swarms were 472 and 2,112 m respectively. On the swarm at Mallinde, which was located closest to the forest, nest site scouts danced for nearer sites than those of the other swarms (Kruskal–Wallis test: χ2 = 10.92, P = 0.0049; Dunn’s multiple comparison: Mallinde vs. Weberstedt (P < 0.05*), Mallinde vs. Craula (P < 0.01**), Craula vs. Weberstedt (P > 0.05)).

Figure 3 Potential dispersal distances of honey bee swarms.

(A) Overall pattern of nest sites found by the experimental swarms set up around the Hainich forest (gray). Blue asterisks depict the positions of the three swarms and dots give the position of advertised nest sites as inferred by decoding the nest site dances of these swarms in Weberstedt (yellow dots), Craula (red dots) and at Mallinde (violet dots). One advertised nest site of each swarm in Craula and at Malinde lay outside the range of the presented map (Map data © Mapbox, © OpenStreetMap). (B) Frequency distribution of potential nest site distances as indicated by dancing bees on the experimental swarms. Solid lines indicate the median and dashed lines give the 95th percentile range of nest site distances.

Discussion

We set out to address a simple question: are there naturally nesting honey bee colonies living in German beech forests? Our results show that the answer is yes, but at low densities. Through the use of beelining technique in the Hainich National Park, where there is no beekeeping activity, we got a picture of the distribution and density of feral colonies in a near-natural beech forest. We found that honey bee colonies were not confined to the forest edges; they were also living deep inside the wood. We inferred a density of around 0.13 colonies per square kilometer. This estimate must be regarded as conservative because we assumed that we would have kept track of any colony in a large area of 2 km around each sampling site. Especially in the core zone of the National Park, there were no opportunities to sample foraging worker bees, so that we might not have sampled the area exhaustively. Further, as we could not exactly locate the actual cavity trees, there might have been more than one bee tree at their inferred locations. Our direct survey of cavity trees in the beech forests of the Swabian Alb corroborated the findings from the Hainich forest that honey bee colonies are regularly inhabiting old tree cavities in beech forests. Our estimates of the lower limit for the colony densities in both woodland areas are similar and are in accordance with the feral colony density estimated in Northern Poland (0.1 colonies/km2) (Oleksa, Gawroński & Tofilski, 2013). However, it is quite low in comparison to the population density of feral honey bee colonies found in the temperate forests of New York State (1.0 colonies/km2) (Seeley, 2007; Seeley et al., 2015).

It is clear that our data present a “snapshot,” without information about the life-histories of individual honey bee colonies. However, we think that they provide some valuable information: The occurrence of feral honey bee colonies in near-natural beech forests is the norm rather than an exception. We found several honey bee nests in both of the two independent woodlands, and in the Hainich forest there were occupied bee trees in both study years. In a pilot beelining tests in a forest area near Würzburg in 2016, we also found feral honey bee colonies.

It is clear that the density of feral colonies in a near-natural deciduous forest in Germany is much lower than the density of beekeeper-managed colonies in rural and urban areas, which is generally between one and three colonies per square kilometer (Jones, 2004). However, when using our estimates to extrapolate the total population size of feral honey bee colonies living in German forest stands older than 80 years, we infer a size of 4,400–5,600 colonies, which we think is a noteworthy number.

The data gathered in the beech forests of the Swabian Alb provide valuable information for future studies and conservation. We hypothesize that the abundance and activity of black woodpeckers is a key factor determining the density of cavities that are large enough for honey bee colonies in managed forests. Beech trees with a diameter at breast height of about 40 cm, which start to be attractive for the woodpecker (Taux, 1976), still have intact sapwood and bark. However, the bird is capable of identifying fungal activity in a tree’s central heartwood and can make cavities in trees that would otherwise lack any microhabitat structures (Zahner, Sikora & Pasinelli, 2012). Interestingly, the black woodpecker reached its present day population size in Central Europe not before the 1950s, probably as a response to the recovering forests (Cuisin, 1985; Blume, 1996). This suggests that protecting large habitat trees in managed forests not only helps charismatic forest birds, but can also promote the occurrence and abundance of feral honey bee colonies.

We found that scout bees of swarms set out around the Hainich forest advertised several nest sites within short distances, mostly outside the forest. On the swarm which was set up closest to the forest edge, scout bees advertised nest sites at closer distances than those of the other swarms, possibly because there were even more suitable nest sites nearby. Prior studies documented similarly short dispersal distances (modal value: 450 m) (Lindauer, 1955; Seeley & Morse, 1977; Seeley, 2010), suggesting that honey bee swarms will always make the most economical decision and move little when suitable nest sites are close by (Seeley, 2010). Our data suggest that there are suitable nesting sites for honey bee colonies in the agro-urban space, and that a beech forest like the Hainich provides ample, but not necessarily better nesting opportunities. Given that the median distance of advertised nest sites in our swarms was only 470 m, the bee trees in the Hainich forest, which were located at a median distance of 2,600 m from the next apiaries, were probably not colonized by swarms from beekeepers’ hives in single dispersal events.

Forest dwelling feral honey bee colonies live in another environment than managed colonies and are subjected to natural selection: they nest in widely spaced tree cavities, no one treats them against parasites, nor are they fed in times of nectar scarcity. There needs to be a certain network of tree cavities for a noteworthy number of feral colonies to persist in any region. This basic requirement seems to be at least partially fulfilled in near-natural or ecologically managed deciduous forests or where there are networks of forest fragments and rural avenues lined with old trees (Oleksa, Gawroński & Tofilski, 2013). It will be interesting to study the life-history traits of feral honey bee colonies in European landscapes and to pose questions such as the following: What is the average longevity of feral colonies and what is the lifetime reproductive success of their queens? Is the number of feral colonies actually limited by parasites in the first place? Or is it the availability of suitable cavities, or resource limitation that keeps feral honey bee colonies at low densities? Understanding and comparing the drivers of the feral honey bee colony density in different woodland areas will also help to answer the question how frequent honey bees would naturally be in temperate Europe.

Irrespective of the fate of individual colonies, the regular nesting of honey bees in woodlands has ecological impacts. Beekeeping activity is not distributed equally across the country but concentrated in rural and urban areas. Managed hives are usually only occasionally moved into the forests (Banaszak, 2009). Consequently, this leads to an uneven distribution of foraging worker honey bees in the landscape, regardless of differences in resource availability among habitats. However, when there are feral colonies living in a forest, the density of foraging workers in that area will increase, with possible consequences for the pollinator community composition of forest dwelling plants (Steffan-Dewenter & Tscharntke, 2000; Mallinger, Gaines-Day & Gratton, 2017; Hung et al., 2018). This is also relevant if the density of feral colonies is low. Based on their colony size and the proportion of foraging workers, it must be assumed that a single honey bee colony sends out up to 5,000 foraging bees daily (Seeley, 1985). When such colonies live in forest fragments surrounded by arable land, they may also contribute to the pollination of nearby crops (Mitchell, Bennett & Gonzalez, 2014; Decocq et al., 2016). Another point is that feral honey bee colonies will directly or indirectly interact with other organisms that live in hollow trees. They certainly compete for tree cavities that are also attractive to other animals (Johnsson, Nilsson & Tjernberg, 1993; Sikora, Schnitt & Kinser, 2016). Further, as they transport nutrients to their nests, honey bee colonies are hotspots of accumulated biomass: individuals can directly be preyed upon and consumed by birds, mammals and other insects (Morse & Nowogrodzki, 1990). And through the drop of organic matter like wax, pollen, or dead bees, honey bee nests indirectly nourish a whole cosmos of organisms that live from organic detritus inside tree cavities. As tree cavities with dead organic matter are generally rare in managed forests, the presence of honey bees could have a positive effect on the abundance of specialized arthropod communities in European beech forests (Möller, 2009; Müller et al., 2014).

Conclusion

Our study showed that honey bee colonies nesting naturally in tree cavities are more common than it is generally assumed. When assessing the impact of honey bees on ecosystems, or when evaluating the need for their conservation, one should differentiate between managed colonies aggregated in apiaries and feral colonies living singly in natural nests. In Europe, A. mellifera is treated as a domesticated animal and feral honey bee colonies lack any legal protection. There is a lot to be investigated about the life of feral honey bees and their interactions with the (forest) environment. However changing the perspective and recognizing that the honey bee can also live as a wild animal is a prerequisite for achieving a full understanding of its biology.

Supplemental Information

Supplemental Information 1 Data on dance circuit duration of waggle dancing bees in relation to flight distances.

Data on dance circuit duration of waggle dancing bees in relation to flight distances as presented by von Frisch for distances between 100 and 6,000 metres. The first linear regression line well describes the relationship for distances up to about 800 metres: circuitduration (s) = 0.0021* distance (m) − 1.4552. Conversely, we used distance (m) = 466.5495* circuitduration (s) − 675.0336 to infer nest site distances from dances with circuit durations up to 3.15 seconds. The second linear regression line well describes the relationship for distances from 900 metres to 6000 metres: circuitduration (s) = 0.0009* distance (m) − 2.4391, and we used distance (m) = 1102.7328* circuitduration (s) − 2666.6256 to infer nest site distances from dances with circuit durations > 3.15 seconds.

Click here for additional data file.

Supplemental Information 2 Detailed picture describing four bee hunts with beelining technique.

When we had problems observing vanishing bearings beneath the dense tree canopy, we used the minimum round trip time to calculate the maximum distance from the feeder to the nest (red transparent circles). Nest site A was predicted to be on the beeline from feeder A1 and inside the maximum nest range obtained at feeder A2. Nest site B was predicted to be at the crossing of the beelines from the feeders B2 and B3, and within the maximum nest range inferred from feeder B4. Nest site C was predicted to be on the beeline from feeder C1 and very close to the feeder C4 where the round trip time was shortest (Note that the predicted nest site for C lies also within the maximum nest ranges inferred from the feeders C2 and C3). Nest site D was predicted based on the beeline from feeder D5 and the maximum nest range inferred from feeder D4 (Map data © Mapbox, © OpenStreetMap).

Click here for additional data file.

Supplemental Information 3 Map of the Swabian Alb.

Map of the 98 inspected cavity trees (blue dots) in the Biosphere Reserve Swabian Alb with the 7 cavity trees occupied by honey bees (red asterisks) (Map data © Mapbox, © OpenStreetMap).

Click here for additional data file.

Supplemental Information 4 Beelining-raw data.

Detailed list of all beelinies obtained in the Hainich National Park. All feeding locations where bees were caught are highlighted yellow and the coordinates, date and time are given. For every marked individual bee we calculated from the round trip time the maximum distance to the nest site. With the flight direction of this individual bee the coordinates to the predicted bee tree are given. Some beelines are highlighted in light grey, they are either missing round trip time or flight direction.

Click here for additional data file.

Supplemental Information 5 Advertised nest sites.

For the three swarms set up around the Hainich we calculated the locations for the potential nest sites inferred from the observed nest site dances. With the circuit duration, waggle run angle and the solar azimuth the nest site coordinates are given. The Number of waggle run circuit for every dance is also given.

Click here for additional data file.

We thank the administration of the Hainich National Park for the possibility of conducting our study in the core zone of the National Park. We are grateful to Luis G. Sikora who kindly provided the GPS coordinates of habitat trees and helped us to find and inspect these trees. We also thank the Deutsche Wildtierstiftung for sharing the habitat tree data and the administration of the Biosphere Reserve Swabian Alb for enabling us to work there. For their valuable comments on the manuscript, we thank Harald Detering, Fabienne Maihoff, Manuel Püttmanns, Sereina Rutschmann and Michael Tölle-Nelken. Last but not least, we thank Jürgen Tautz and Ingolf Steffan-Dewenter for their suggestions how to improve this paper and for supporting this study.

Additional Information and Declarations

Competing Interests

Author Contributions

Data Availability

The authors declare that they have no competing interests.

Patrick Laurenz Kohl conceived and designed the experiments, performed the experiments, analyzed the data, contributed reagents/materials/analysis tools, prepared figures and/or tables, authored or reviewed drafts of the paper, approved the final draft.

Benjamin Rutschmann conceived and designed the experiments, performed the experiments, analyzed the data, contributed reagents/materials/analysis tools, prepared figures and/or tables, authored or reviewed drafts of the paper, approved the final draft.

The following information was supplied regarding data availability:

The raw data is provided in Supplemental Dataset Files.

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
