# Peer review of "The neglected bee trees: European beech forests as a home for feral honey bee colonies"

_PeerJ, doi:10.7717/peerj.4602_

## Round 0.1 · original submission · Minor Revisions

I think that the manuscript is robust and well written. It needs only few revisions prior to be published.

·

Basic reporting

1. Basic reporting. Excellent. Authors introduce well the aims and rationale for their investigation: to initiate the study of wild honey bee colonies in European hardwood forests by determining minimum estimates of colony density. They also explain that although Apis mellifera is native to Europe, there is almost nothing known about the wild colonies of this species within Europe, hence this is a path-breaking investigation. The writing is clear and precise and grammatically correct.

Experimental design

2. Experimental design: Clear. The authors have adapted methods used in North America to locate honey bee colonies living in forests, and they have applied these methods to their study sites with skill. Also, they have set up artificial swarms properly and taken meaningful data from them by reading the waggle dances of nest-site scouts. They did well to feed these artificial swarms with sugar water while they were house hunting, to eliminate the possibility that the dances they observed were advertising flower patches, not potential nest sites.

Validity of the findings

3. Results and Discussion: Generally excellent, with super graphics, but I have some suggestions for improvement.

Line 67: The references cited here are Seeley and Morse 1976 and Seeley 1985. It would be better to cite Visscher and Seeley 1982, Seeley 2017, and Seeley et al. 2015, for these three papers provide clearer evidence that in nature colonies nest in widely spaced tree cavities.

Line 69: Besides being smaller and swarming more frequently, wild colonies have less brood; these are all things that make them less susceptible to varroa mites.

Line 77-78: Recent work, about to be submitted, has revealed that the honey bees living in the Arnot Forest possess multiple behavioral mechanisms of resistance to varroa. These include intense hygienic behavior (99% of infested cells are cleaned out within 48 h) and good biting behavior (ca. 40% of phoretic mites are killed by having legs bitten off). So I recommend changing this sentence, and giving as the reference the following:
"Peck and Seeley, personal communication."

Line 85: Yes, it is very good to begin to gather information from European forests.

Line 95-97: I agree with the view that the conclusions by Moritz et al. 2007 and Jaffe et al. 2010 (that wild colonies were not present in their study sites) are weakly supported. The method used in these studies to estimate colony density require making many, many assumptions (e.g. about drone and queen flight distances, etc) so the estimates of colony density that were reported are very imprecise.

Line 156: Replace "are quite site fidel" with "have high site fidelity."

Line 216: Replace "swiped" with "swept".

Fig. 4, right side: It would be good to relate the results shown here to the results on dispersal distance reported in Seeley and Morse 1977. In this study, the modal distance to prospective nest sites was 450 m.

Line 314. It is stated here that Visscher and Seeley (1982) reported a density of wild colonies of only 0.67/km2. This is true; they found 11 wild colonies within the 16 km2 Arnot Forest. However, they should have taken into account the fact that not all 16 km2 of this forest were searched. Actually, only about 67% of the forest was searched, so a better estimate of the density of wild colonies in this forest is 1.00 colonies per km2. This estimate is supported by the findings of later studies; they have consistently yielded estimates of colony density of 1 colony/km2. See Seeley (2007) and Seeley et al. (2015).

In the references, all the references to Seeley should indicate Seeley, T.D. At present, it sometimes says Seeley, T. and sometimes Seeley T.D., but they are the same person, and he goes by T.D. Seeley.

Additional comments

You have done an excellent job of collecting your data and in presenting your findings.

Your graphics are first rate, and I like the use of the yellow footprints to denote the track along which you moved during your bee hunting.

Thank you for conducting this pioneering work!

Reviewer 2 ·

Basic reporting

Kohl & Rutschmann led an excellent study that characterizes the densities of feral honey bees in forested areas in Germany. In the introduction, the authors give an informative and well-written background section with detailed citations, which gives the reader the necessary information to understand the relevance of the study and the need for the data that they collected. The language used and structure of the paper is clear.

The goal of the study was to estimate the density of feral honey bee colonies in forested areas using two different methods and understanding more about the role of swarming in the dynamics of these feral colonies. The authors clearly report their results about the 3 aspects of the study.

We only have minor comments that may help clarify some points throughout.

Experimental design

Robust experimental design. Methods are explained in a clear manner for the most part. Please see comments below about some aspects of the figures requiring clarification.

Validity of the findings

Conclusions are well stated. Results are extremely valuable as baseline data for monitoring feral European honey bee colonies in their native range. This study challenges the notion that feral honey bees disappeared in Europe after the introduction of varroa mites.

Additional comments

Some minor comments:

* L13-14: Authors should note and cite the study by Thompson et al 2014 that report virus levels on feral colonies in the UK.

* L19-20: This sentence is unclear. We suggest: “Further, we estimated flight distances of scout bees from swarms after decoding dances near forest edges”.

* L92-94: Microsatellite genotyping for colony density estimation requires markers that are unlinked.

* L96-98: Besides the possible lack of ability to detect low colony densities, the problem of using genetics for these estimates is that managed and feral bees are not necessarily isolated populations. Therefore, the power to detect colony densities as if these were separate “populations” is probably low.

* L103: What do you mean by legal status?

* L156: Consider rephrasing to “foraging honey bees exhibit site fidelity”

* Figure 2. It is unclear why figure 1 and 2 were included. Please clarify these two versions of the results of beelining in Hainich.

* L300 - Please clarify low densities compared to what.

Margarita López-Uribe
Chauncy Hinshaw

---

## Round 0.2 · accepted · Accept

I think that this manuscript is now ready to be published on PeerJ

#